# Avatar-Based Strategies for Breast Cancer Patients: A Systematic Review

**DOI:** 10.3390/cancers15164031

**Published:** 2023-08-09

**Authors:** Irene Rodríguez-Guidonet, Paula Andrade-Pino, Carlos Monfort-Vinuesa, Esther Rincon

**Affiliations:** 1Psycho-Technology Lab, Universidad San Pablo-CEU, CEU Universities, 28005 Madrid, Spain; 2Departamento de Psicología y Pedagogía, Facultad de Medicina, Universidad San Pablo-CEU, CEU Universities, Urbanización Montepríncipe, 28005 Madrid, Spain; 3Departamento de Medicina Interna, HM Hospital, Universidad San Pablo-CEU, CEU Universities, 28005 Madrid, Spain

**Keywords:** breast cancer, quality of life, well-being, avatar

## Abstract

**Simple Summary:**

Breast cancer is the most commonly diagnosed cancer worldwide, and its burden has been rising over the past decades. As Internet or smartphone applications can be useful in addressing psychological burdens in cancer patients, these new tools represent an innovative research topic. Furthermore, among those new technological strategies, avatar-based treatments are starting to be considered a promising way to provide educational and psychological support to breast cancer (BC) patients. However, to date, no study has investigated the potential benefits that avatar-based technology could represent for improving the quality of life (QoL) and well-being of BC patients.

**Abstract:**

There is a lack of studies to determine if avatar-based protocols could be considered an efficient and accurate strategy to improve psychological well-being in oncology patients, even though it represents a growing field of research. To the best of our knowledge, this is the first systematic review addressing the effectiveness of avatar-based treatments to enhance quality of life (QoL) and psychological well-being in breast cancer patients. The purpose of this study was to review the scientific literature of those studies involving avatar-based technology and breast cancer patients in order to answer the following questions. (1) Are avatar-based strategies useful to im-prove QoL and psychological well-being (anxiety and depression symptoms) in breast cancer patients? (2) Which is the best way to develop avatar-based protocols for breast cancer patients? We conducted a systematic review of the peer-reviewed literature from EBSCO, Ovid, PubMed, Scopus, and Web of Science (WOS), following the PRISMA statements and using “avatar + breast cancer” or “avatar + cancer” as keywords. Studies which were published in either English or Spanish and which addressed QoL and psychological well-being in breast cancer patients were reviewed. The results will contribute to developing innovative avatar-based strategies focused on breast cancer patients.

## 1. Introduction

Breast cancer (BC) is the most commonly diagnosed cancer worldwide, and its burden has been rising over the past decades, becoming the second leading cause of death [1]. Over 2.3 million new cases and 685,000 deaths from breast cancer occurred in 2020, being the most frequent cause of death in women under 55 years old and accounting for 15% of all cancer cases [2]. By 2040, the breast cancer burden is predicted to increase to over 3 million new cases and 1 million deaths every year [3]. It represents the most common malignancy diagnosed in women in the USA, with 281,550 new cases of invasive breast cancer and 49,290 new cases of noninvasive breast cancer diagnosed per year [4]; additionally, the frequency of this diagnosis is increasing in low- and middle-income countries [5]. 

Despite the advances in cancer treatment, BC patients experience notable psychological consequences that impact their quality of life (QoL) and recovery after care [6,7,8]. This is associated with the illness experience that changes individual lifestyle, activities, and reshapes future plans [1]. Survivors may also experience depression and anxiety symptoms [6], as well as fatigue, pain, and distortions in intimacy (loss of pleasure, sexual desire, and womanhood) [9,10]. Fertility disorders after hormone therapy, as well as body image distortions following breast surgery, have also been mentioned as challenges that women suffering BC must face on a daily basis [6,10]. Other difficulties confronted by BC patients include alterations in their physical appearance such as loss or disfigurement of one or both breasts, hair loss (possibly due to chemotherapy) and/or surgery scars, weight gain from hormone therapy, or early menopause [11,12]. These side effects could impact both body image and self-perception, considering body image is related to multiple variables, such as identity, self-esteem, attractiveness, social relationships, and sexual functioning [10,13]. In the same vein, the literature reports that, in cancer patients, both body image and dissatisfaction with the body have a considerable impact on mental health, stress levels, life satisfaction, self-esteem, the conception of physical health, and, therefore, quality of life [14]. These aspects have been specifically evidenced in women survivors of breast cancer, where the affectation of body image has a greater impact on psychological distress [15].

The oncological diagnosis affects women in the prime of their lives, where they are often in the middle of their careers or busy raising their children, which affects their ability to cope with the life-threatening disease [16]. All the physical and psychological symptoms resulting from treatments may seriously impair women’s daily life routines, and consequently their QoL. In fact, a BC patient’s lifestyle has been stated as having a similar effect on women´s QoL as the usual clinicopathological risk factors [5], evidencing the importance of patients’ QoL as a significant factor for the prognosis of the consequences associated with cancer [17]. In this regard, QoL has been defined as a multidimensional construct that is dynamic over time and includes a variety of domains such as physical, functional, emotional, spiritual, economic, and social [18].

Considered as a dynamic construct, QoL in BC patients may decrease after diagnosis, as well as during the treatment and survival periods [19], due to the disease itself, which requires significant mental and physical adjustment and undoubtedly affects QoL [20]. BC survivors experience a general decrease in QoL compared to the general population [21]. Thus, a diagnosis such as cancer is seen as a traumatic stressor that could have a negative impact on the various areas of functioning of the person, such as the ability to work or develop social relationships, depending on the presence of negative mood [6]. QoL provides significant data that should be taken into consideration for cancer-related prognoses [20].

In BC, social support is essential in building resilience and improving quality of life [22], and significantly influences disease adjustment [23,24] by reducing distress [25,26], depression [27], and risk of recurrence [28,29]. However, not all types of social support are effective; it must be supportive, positive, and characterized by interactions that promote affection and respond to the needs of the patient [30].

Considering the negative impact of cancer in a person, it is recommended that BC women receive effective social and professional support to be able to cope with all the physical and psychosocial consequences of cancer diagnosis. However, the majority of cancer survivors experienced the feeling of not having an effective network to turn to in the face of problems that arise after diagnosis or perceived the absence of psychological support to deal with the diagnosis [19,31].

Based on the lack of perceived support reported by BC patients, alongside the traditional psychological procedures, internet-based psychotherapy may be considered as an alternative way of providing psychosocial care [1]. It allows mental health professionals to reach patients limited by geographical distance (e.g., living in rural areas) while including digital health technology for symptom monitoring such as health apps [1]. The Internet, as a tool for treatment, is an opportunity to promote the dissemination of psychological interventions [1]. As such, patients could gain access to psychotherapists, thereby overcoming barriers (e.g., geographical distance and limited access to treatment) [1]. 

Thus, technology has been adapted to the various needs derived from the health field, and eHealth has incorporated the applications of new information and communication technologies (ICT). It not only contributes as a management tool, but also helps in health promotion [31], since its objectives are to respond to the affectations of patients and thus favor their integral development [32], as well as to facilitate active participation of the patient in their treatment, favoring the organization of their daily life and mobilizing a modification in the self-perception of their life [33]. The relevance of new technologies in eHealth and the relationship with QoL has been previously reported [6], considering that they not only contribute as management tools but also help in health promotion [18]. Several studies reported that eHealth is operational and productive, demonstrating that patients present high results in both acceptance and satisfaction [6]. Positive Technology can guide the development of eHealth tools that foster patient motivation towards prevention and self-care by improving their autonomy and empowerment in their health care process, being adapted to the specific needs and conditions of its patient [34]. In BC patients, several benefits of using eHealth strategies have been mentioned [6]. 

Among eHealth strategies, a promising resource for psychotherapy is represented by avatars, which are digital representations of patients that interact with professionals and/or peers, in a virtual environment [1]. Avatars are customized by users, and generally, such customization represents aspects of their personality, self, and physical appearance that could be analyzed in the context of psychological assessment [33,35,36]. In this regard, avatars are digital representations of the self that are used in the context of the Inter-net within digital environments, encouraging not only social interactions and interaction in a virtual environment among users, but also the adoption of adaptive or healthy behaviors [37,38].

Among the main advantages of using avatars to approach patients, in the field of cyberpsychology, aspects that converge in an opportunity are relevant, such as the relationship that the avatar has with the identity of the patient; the positive impact it entails in the modification of behavior; serving as a resource that enables the analysis of the patient’s internal experience including personalization and the representation they make of their avatar; the support that emerges at the level of social relationships; and the sense of belonging to a group and the supports that arise from it, where it is possible to favor responsibility in personal health care from the positive influence provided by the group experience [33]. The role of avatars in e-Mental health interventions has also been investigated, wherein some therapeutic functions have been described, such as the support of a therapeutic relationship through a virtual presence, the reduction of communication barriers, and anonymity promoting treatment-seeking behaviors [39].

In addition, other positive aspects are related to the diversity of their application in a heterogeneity of pathologies [40], their ability to train emotions in patients with schizophrenia or related disorders [38]; as well as in major depressive disorder [41]; in anxiety disorders [42]; or in autism spectrum disorder [43], among others. 

Therefore, the use of virtual experiences through avatars in virtual environments as a therapeutic tool is evident and is gaining ground in the interest of academics and health professionals [44]. 

In this regard, [45] investigated the effectiveness of creating an avatar as a self-representation in a virtual environment in order to avoid unhealthy behaviors that arise from the difference between the idealized image of the body and reality, seeking to motivate people to achieve their ideal body image and maintain a healthy lifestyle. They concluded that creating avatars to reflect the ideal self can be an effective tool to motivate prevention and self-preservation. Likewise, in chronic diseases such as cancer, variables such as knowledge, self-care behaviors, and improved QoL have been favored, demonstrating promising results in terms of effectiveness for patient education (improving knowledge and self-care behaviors in cancer patients) [46]. Avatar-based strategies have been also mentioned as an effective tool to improve patients’ mental health and chronic care [38]. 

When patients were asked about their user experience, they reported a high level of user satisfaction after using avatar-based tools [38]. Similarly, when BC patients were questioned about their attitudes toward internet-based psychotherapy, including avatar therapy, the results showed a positive attitude towards this new technological tool [1]. In previous studies, the avatar has been mentioned as useful to communicate meaning about emotions and thoughts and to express facets of the self [47], as well as benefiting the recognition of personal resources and promoting the differentiation between the affectation of the disease, as in “I patient”, with the other aspects that encompass its identity, that is, other possible I’s [48]. From this point of view, the use of avatars in BC patients could be a useful tool at the therapeutic level, to address the psychological affectations derived from the disease, taking into consideration the psychotherapeutic experiences at the group level, where the socialization of the gender role, objectified body awareness, and quality of life have been addressed [10].

Regarding the disadvantages that should be considered in the use of ICT and avatars in the health field with patients, new communication barriers produced by the use of ava-tars, the extra challenge that some clients may have to overcome in learning how to use avatars, ethical concerns (i.e., privacy or the replacement of face-to-face support by digital technologies) [49], or the initial cost have been mentioned [50]. It has also been considered that, in the initial phase of the disease, known as “blackout”, according to the patient participation model (PHE) [51], an emotional, cognitive, and behavioral “blackout” would be common among patients after a critical event such as receiving a new diagnosis or worsening of the disease, among others. Therefore, patients may not be receptive to starting treatments because they must effectively cope with the disease itself, understanding that they are in the initial stages of connecting with their new state of health. Therefore, it could be complex that, in addition to dealing with the disease, they initially contemplate the benefits that could be provided by the support of a virtual world [33], although in the same way, the use of ICT could be beneficial in this phase to modify the emotional state of the patient and enable them to regain control [34].

Despite the aforementioned progress in addressing avatar-based protocols’ benefits for chronic patients, little knowledge has been provided about their efficacy in BC patients, especially concerning physical and psychosocial side effects. Considering that avatar-based strategies could represent a useful tool to help in BC patients’ treatment and recovery processes, their efficacy is worth analyzing based on recent and peer-reviewed scientific literature. As such, this study aimed to answer the two following questions. (1) Are avatar-based strategies useful to improve QoL and psychological well-being (anxiety and depressive symptoms) in BC patients? (2) Which could be the most successful path to develop avatar-based strategies involving BC patients?

## 2. Materials and Methods

### 2.1. General Description

A systematic search strategy was implemented in November 2022 to find all the relevant studies involving the use of avatar-based strategies in BC patients. It was performed and reported using the Preferred Reporting Items for Systematic Reviews and Meta-Analyses (PRISMA) Statement (see study protocol in Appendix A) [52]. The protocol was registered with the PROSPERO International Prospective Register of Systematic Reviews (CRD42023399369).

### 2.2. Selection Criteria 

Inclusion criteria: The study papers were considered relevant if they were journal articles involving avatar-based protocols to enhance QoL and psychological well-being (anxiety and depressive symptoms) in BC patients. Studies should be published in either English or Spanish, over the past decade (between 2012 and November 2022), providing specific outcomes (quantitative results).

Exclusion criteria: Studies including the use of avatar-based strategies that did not involve BC patients or that did not measure QoL or psychological well-being (anxiety and depressive symptoms) were discarded. Protocols with unpublished results, narrative re-views, no journal articles (conference proceedings, book chapters, or theses), or published in a language other than English or Spanish were also excluded.

### 2.3. Outcomes

The primary outcomes were the type of avatar-based related treatment developed, if the improvement in mental health was measured through an empirically validated questionnaire, and if the avatar-based strategy involved was useful to enhance QoL and psychological well-being (anxiety and depressive symptoms) in BC patients. The secondary outcomes were the main advantages and disadvantages of the training provided, as well as the patients’ satisfaction levels after using the avatars.

### 2.4. Search Methodology 

A comprehensive search was carried out in EBSCO (Academic Search Complete, CI-NAHL Plus with Full Text, Communication Source, eBook Collection, E-Journals, ERIC, Fuente Academica Premier, Humanities International Complete, MEDLINE, MLA Directo-ry of Periodicals, MLA International Bibliography, OpenDissertations, PSICODOC, Psychology and Behavioral Sciences Collection, PsycInfo), Ovid, PubMed, Scopus, and WOS (Web of Science Core Collection) from inception until November 2022 using the following keywords: “avatar” + “cancer”; and “avatar” + “breast cancer”. The detailed search strategies used in all databases are provided in Appendix A. All original research articles were retrieved for examination, and a search library was created using Ref-Works©, a bibliography management program. 

### 2.5. Data Collection and Analysis 

Two authors (I.R.-G. and P.A.-P.) independently evaluated and reviewed for completeness all titles and abstracts following three phases: first, the titles of the records were assessed, followed by their abstracts; finally, if after reading the titles and the abstracts, a reviewer considered that a reference was relevant, the full text of the paper was extracted. After this, Cohen kappa scores were calculated to measure the inter-rater agreement between the two researchers (I.R.-G. and P.A.-P.). 

The interpretation of the Cohen kappa coefficient was calculated using SPSS version 27 (IBM Corp.) and was based on the categories developed by Douglas Altman [53]: 0.00–0.20 (poor), 0.21–0.40 (fair), 0.41–0.60 (moderate), 0.61–0.80 (good), and 0.81–1.00 (very good). In case of discrepancies, a third author was consulted (C.M.-V.). Cross-checking was carried out to identify any inaccuracies or oversights (E.R.). Any other discrepancies were resolved among the core team with the involvement of the broader research team when necessary. 

### 2.6. Data Extraction and Management 

We extracted data based on (1) publication year, (2) country, (3) study design, (4) study aim, (5) sample size (and mean participants’ age), (6) if all the participants included were breast cancer patients and cancer stage, (7) if patients were randomized, (8) training using avatar, (9) QoL and well-being measured (yes/no instrument used), (10) whether QoL/ psychological well-being could be improved, (11) the main advantages/disadvantages, and (12) patients’ satisfaction.

### 2.7. Quality of Studies Included 

Given the variety of the research designs, the quality of included studies were ap-praised using the Mixed Methods Appraisal Tool (MMAT) developed in 2006 [54] and updated in 2018 [55]. The overall scores with the highest values indicated a lower quality of included studies (see Appendix A). One author (C.M.-V.) independently extracted data on outcomes from all studies. Data were reviewed for completeness by one reviewer (E.R.). 

### 2.8. Statistical Analysis 

Data were pooled using the program SPSS v. 27 (IBM Corp., Armonk, NY, USA), which allowed an analysis of frequencies (percentages), as well as means.

## 3. Results

### 3.1. Study Selection and Inclusion

A total of 618 records were included in RefWorks© through the electronic database search. After removing 440 duplicates, another 12 studies were discarded for not complying with the inclusion criteria (no journal articles). Subsequently, 166 records were evaluated based on the title and abstract. Of those, 157 were discarded because they did not meet the inclusion criteria. Therefore, nine papers were selected for a full text reading; six out of these [1,56,57,58,59,60] were excluded for various reasons (see Appendix A). A total of three publications were finally included [61,62,63]. The Cohen kappa showed a substantial level of agreement, and it was categorized as “good” (κ = 0.64) (range 0.61–0.80) based on the categories developed by Altman [53]. A PRISMA flow diagram [52] is provided in Figure 1. All chosen studies were deemed to be of sufficient quality to contribute equally to the thematic synthesis. 

### 3.2. General Characteristics of the Studies Included

Regarding points 1 (year of publication), 2 (country of the study), and 3 (study de-sign), the following results were extracted (Table 1): the three selected studies were published between 2016 (n = 1) [62] and 2021 (33.33%) (n = 1) [61]. The majority of the studies were conducted in the United States (n = 2; 66.7%) [61,62]. The remaining paper was published in Italy (n = 1; 33.33.7%) [63]. The included studies followed a mixed method (n = 2; 66.7%) [61,62] or quantitative method approach (n = 1; 33.33%) [63] (Table 1). 

Addressing points 4 (study aim), 5 (sample size and mean participants’ age), 6 (if all the participants included were breast cancer patients (yes/no) and cancer stage), and 7 (if patients were randomized (yes/no)), the following results were extracted. The objectives of the studies were extremely varied (Table 2). In one study (33.33%), a feasibility trial was carried out in order to establish the acceptability of an avatar-facilitated life review intervention for ambulatory patients with cancer [61]. Another study (33.33%) evaluated the usability, feasibility, and acceptability of TOLF, an educational and behavioral health IT system, as well as the usability, feasibility, acceptability, and efficacy of mHealth self-care interventions for lymphedema symptoms among BC survivors [62]. Finally, the third included record consisted of an exploratory study to examine how breast cancer affects ava-tar-conveyed multiple representations and determine if attitudes toward avatars, reconceptualized as attitudes toward representation of one’s multiple selves, were related to anxiety and depression symptoms [63]. 

Sample sizes ranged from 12 [61] to 355 [62] patients involved. There were two studies (66.66%) that did not provide the mean participant age [61,62,63]. A total of 389 pa-tients were involved in the three studies analyzed, with the majority of them being only BC patients (n = 2; 66.7%) [62,63]. Just one record (33.33%) targeted patients with different cancer types [61], involving one BC patient, along with other cancer patients: genitourinary (33.3%), gastrointestinal (25%), myeloma (8.3%), lung (8.3%), neuroendocrine (8.3%), and sarcoma (8.3%)] (Table 2).

#### Assessment of Methodological Quality of Included Studies

A vast diversity was found in the design of the studies, as well as in the statistical methods used with variety in the presentation of the results obtained (see Appendix A).

### 3.3. Primary Outcomes

In relation to points 8 (training using Avatar), 9 (QoL and well-being measured and instrument used), and 10 (usefulness to improve QoL and/or psychological well-being), the following results were extracted (Table 3). 

#### 3.3.1. Training Using Avatar

In the first study included [61], avatars were used for life review in the therapeutic model (outpatient setting), consisting of one session using VoicingHan, an avatar-facilitated life review intervention that utilized a Perception Neuron Pro Motion Capture device and a Logitech^®^ wireless headset system, which synchronized voice, gestures, and movements of participants onto an avatar-based virtual environment. It includes five different environments that the patient was able to choose between, as well as 64 personalized avatar options, including four male and four female avatars presented at various stages of development (child, adolescent, adult, and elderly). The process began at the child-stage avatar, progressing through adolescence, adulthood, and old age, asking the patients questions related to the stage in which they found themselves and encouraging them to review their personal histories and to reflect on their past, present, and future. 

The-Optimal-Lymph-Flow (TOLF) program is a patient-centered web- and mobile-based educational and behavioral program targeting compromised lymphatic systems and focusing on self-care strategies to help patients to learn self-care strategies [62]. Patients could monitor their lymphedema symptom experience virtually and remotely. It al-so gave recommendations for self-care strategies to encourage the development of self-care skills, in which users initiate and perform activities to prevent, relieve, or decrease lymphedema symptom occurrence to manage their symptoms, which has been shown to be effective in reducing the risk of lymphedema and relieving symptoms. TOLF is a digital strategy in which breast cancer survivors initiate and perform activities to prevent, relieve, or decrease lymphedema symptom occurrence (i.e., the number and severity of symptoms) and distress as well as improve their QOL. The pilot trial was focused on how TOLF may enhance patients’ self-care for lymphedema symptom management. In this study, breast cancer patients were able to learn and follow all the exercises suggested through avatar video simulations. As such, involving 30 breast cancer survivors, the researchers studied the usability and feasibility of TOLF, highlighting that patients loved the web-based program, especially the videos using the avatar technology to show the complex lymphatic system and exemplifying each practice, as well as demonstrating step-by-step instructions for each exercise [62].

In the third included study [63], the researchers asked the participants to create avatars using a free mobile app called Profile Avatar Maker 2 (Google Commerce Ltd., Dublin, Ireland), which allowed the user to choose several aspects of their avatars, such as gender, face physiognomy, emotional expression, hair, eyes, clothes and accessories, and background. Participants received detailed instructions to create three different avatars that had to resemble their actual self (AS), ideal self (IS), and self with cancer (Table 3). 

#### 3.3.2. Measured QoL and Well-Being and the Instrument Used

Only one (33.33%) [61] out of the three included studies assessed the QoL directly. The first included study [61] assessed the patients’ QoL and spiritual well-being using the European Organization for Research and Treatment of Cancer-Quality of Life Questionnaire Core 30 (EORTC QLQ-C30) [64] and the Spiritual Well-Being Scale (FACIT-Sp) [65] at baseline versus follow-up (post-test and one month later). The EORTC QLQ-C30 [64] presents a 30-item generic cancer questionnaire, which contains five function scales (physical, role, emotional, cognitive, and social), another global health scale, and three multi-item symptom scales (fatigue, nausea/vomiting, and pain), as well as six single-item scales (dyspnea, sleep, appetite, constipation, diarrhea, and financial hardship due to ill-ness). The FACIT-Sp [65] has two subscales, which measure, on the one hand, the sense of meaning and peace, and, on the other hand, the role of faith in the disease. Likewise, the questionnaire yields an overall score for spiritual well-being [66]. The two other studies [62,63] did not assess QoL symptoms directly.

TOLF [62] hosted an electronic version of major clinical and research assessment instruments, including demographic and clinical information, as well as the Breast Cancer and Lymphedema Symptom Experience Index (BCLE-SEI), a five-point Likert-type self-report instrument consisting of two parts evaluating the occurrence of and distress from lymphedema symptoms, including the negative impact and suffering evoked by an individual’s experience of lymphedema symptoms; as well as daily living, function, social impact, sleep disturbance, sexuality, emotional/psychological distress, and self-perception [62]. It measured lymphedema symptoms as well as the secondary outcomes of symptom distress/QOL related to pain and symptoms [62]. The third included study [63] focused on clinically relevant states, specifically depression through the Depression Patient Health Questionnaire-9 (PHQ-9) [67] and anxiety through the Anxiety Generalized Anxiety Disorder-7 (GAD-7) [68].

After the creation of the avatars [63], participants filled in three questionnaires, one about their attitudes toward their avatars (ordered randomly) and the other two on de-pression (PHQ-9) [67] and anxiety (GAD-7) [68]. The first questionnaire was a seven-point Likert scale asking to evaluate each avatar regarding its attractiveness, its self-representativeness, the difficulty experienced in creating it, and the emotional state elicited by each avatar in its intensity, pleasantness, and dominance [63]. 

The other variables measured in the studies were patient adherence, recruitment rate, acceptability and comfort of the study procedure, patients’ perceived benefits, symptom burden [61], lymphedema symptom experience, usability, perceptions of information and system quality [62], and attitudes and emotions toward the participants’ avatars [63]. Additionally, information and exercises encouraging users to build self-care skills for their lymphedema symptom were provided, which improved its management and, consequently, improved patient QoL [62] (Table 3). 

#### 3.3.3. Usefulness to Improve QoL and/or Psychological Well-Being

Regarding the effectiveness of avatar-based strategies to improve QoL and/or psychological well-being, the results showed that there were no significant pretest/posttest differences for health-related QoL or spiritual well-being of the patients, and total ESAS scores improved for 6 out of 11 patients included [61]. TOLF helped to improve patients’ QoL and to achieve optimal limb volume difference and lymph fluid level [62]. At 12 weeks post-intervention, in terms of symptom distress/QOL, participants reported that pain had significantly lower interference with their enjoyment of life and less negative impact on the emotion of frustration [62]. Moreover, the daily 5-minute routine avatar simulation video of lymphatic exercises provided a “unique way of helping breast cancer survivors to establish their own self-care routine by following the video” [62]. The results showed that TOLF had significantly positive effects on less pain (*p* = 0.031), less soreness (*p* = 0.021), less aching (*p* = 0.024), less tenderness (*p* = 0.039), fewer numbers of lymphedema symptoms (*p* = 0.003), and improved symptom distress (*p* = 0.000) at 12 weeks after intervention [62] (Table 3).

### 3.4. Secondary Outcomes

#### 3.4.1. Main Advantages and Disadvantages

In the first included study [61], the feasibility, that is, the capacity of the study to carry out the life review intervention with avatars, was estimated based on the indicators of patient adherence, recruitment rate, acceptability, and comfort of the procedures, as well as the patients’ perceived benefits (Table 4). On the other hand, the feasibility of the evaluation of health-related QoL, the effects in relation to spirituality, and the set of symptoms was only exploratory. The final results showed that the patient recruitment rate was 71% and the adherence rate was 92%; the acceptability and comfort of the study procedures were high, with a completion rate relative to avatar intervention of more than 90%; finally, the results suggested potential benefit for patients from a single session. 

Concerning the advantages of the use of avatars in virtual reality (VR) [61], they facilitated navigation in these virtual environments, allowing patients to use gestures and movements, which may enhance patient verbal communication and engagement by reducing communication barriers, providing anonymity, and promoting the expression of patient identity.

TOLF [62] passed a preliminary heuristic evaluation from 15 usability experts before it was tested, in which 90% of participants reported having no usability problems, and feedback about improvement aspects was taken into consideration to refine the system. It is the only one of the three selected studies whose avatar-related technology (in this case, an app and website) was patient-targeted, and it can be downloaded onto computers, lap-tops, tablets, or other electronic devices and smartphones (available both on Android and iOS platforms). 

Its self-care program features innovative, safe, feasible, user-friendly, and easy-to-integrate self-care strategies, with information regarding lymphedema symptom management being considered clear, easy to understand, high quality, and empowering. It has different sections, including Instruction (health information and information on situational self-care strategies), Daily (lymphatic) exercise (by watching and following avatar simulation videos), Friendly reminder (alarm, calendar), and Symptom Update (evaluation and monitoring of symptoms). To prevent technical skill barriers to access TOLF, re-searchers helped BC patients who had any questions or needs regarding the setup or navigation of the system. Each participant was given a user manual with a list of tasks to navigate the system. 

Participants were required to find the information and videos listed in the user manual. Participants had ongoing access to the program during the 12-week study period to review the material as needed using their own computers, laptops, iPads, or other electronic tablets or smartphones [62].

Avatars created were also used as an innovative way to show how the patients’ self-representations may vary between actual self (AS), ideal self (IS), and self with cancer [63]. Therefore, the experience of a chronic illness (a breast cancer diagnosis) was used as an independent source of information to represent a facet of the self [63]. During the process, participants had continual access to the researchers’ support if they encountered any difficulty in using the platform and/or creating the avatars, while their privacy was pre-served throughout the process by providing them with specific codes [63].

The main technological functionalities, considered as key strengths of the avatar-related technological devices were the possibility to synchronize voice, gestures, and movements onto an avatar in a virtual environment, real-time view of each motion capture session [61], educational information, exercise videos, users’ symptom self-evaluation, alarm system for reminders [62], and the creation of different avatars by personalizing their physical characteristics [63].

In regard to the limitations of the studies, it became relevant that, since the intervention required sufficient space for the movements of the patients, it could be more difficult for centers that cannot guarantee this [61]. Likewise, when using special equipment, highly trained human support is required to provide personalized counseling to patients, which can be a major obstacle in institutions that do not have the resources to carry it out [61]. Furthermore, it was considered within the study that there could be a selection bias that favored the high adherence and acceptability rates, based on the fact that the participants were part of the care center and therefore known by this team [61]. 

Among other weaknesses of the included studies, the small samples of the studies and the lack of control groups to compare the results can be cited [61,62,63]. Some selection bias (due to the previous knowledge of the participants), which could have favored the high adherence and acceptability rates, was also mentioned [61]. Likewise, participants were similar in terms of education and familiarity with Internet use [62], which can be addressed via a further usability test of TOLF with a more varied population. Finally [63], there was a lack of measurement of some relevant variables: the avatars’ characteristics, which were impossible to objectively measure due to the simplicity of the avatar creation platform, and the ability and satisfaction of the patients regarding the creation of their avatars (Table 4).

#### 3.4.2. Patients’ Satisfaction 

Only two studies [61,62] reported information on the level of patient satisfaction, which in both cases was high (Table 4). Specifically, the acceptability and perceived benefits in patients in [61] were evaluated through the application of a survey (five-question Likert), resulting in strong, positive evaluations, with an average score of 4.56/5. They reported high acceptability for the intervention, with a completion rate of more than 90% and all of them agreeing or strongly agreeing to participate in the avatar session again, as well as recommending it to others. In conclusion, participants evaluated the experience as beneficial [61]. Satisfaction could be also perceived after TOLF use, as all the participants agreed that the information provided by TOLF regarding lymphedema symptom management was clear, easy to understand, high quality, and empowering [62]. Participants loved the fact that patients can access TOLF remotely and at any time to learn about lymphedema, symptoms, and self-care strategies at their own pace. Considering the usability and feasibility study, patients reported loving avatar simulation videos because they helped them to understand the lymphatic system and learn daily lymphatic exercises. The majority of participants, 96.6%, strongly agreed that the system was easy to use and effective in helping them learn about lymphedema, symptoms, and self-care strategies, while 90% rated the system as having no usability problems.

## 4. Discussion

Avatars are common components of virtual reality programs. They originated for gaming purposes [50], but their use has been spread into other fields, such as in healthcare, where they are applied in a wide range of pathologies and achieve positive outcomes. Patients who have used avatars in therapeutic settings seem to experience improved physical and psychosocial aspects [33,38,39,46,48,50,69]. These digital characters can be adjusted to match cultural, social, and individual preferences, enhancing the overall experience for users [50,69].

There is a paucity of research that examines the use of avatar-related strategies to im-prove the QoL and psychological well-being of breast cancer patients. The examined studies suggest that avatar-based solutions could improve the psychosocial consequences of cancer treatments, but further research is needed in order to make a significant statement. 

Although no QoL or psychological well-being improvement was found, it is important to mention that some aspects of the QoL underwent improvements after using avatars, specifically, through the Life Review [61]. Similarly, there was an increase in the FACIT-Sp scores (spiritual well-being) in more than half of the patients a month later, suggesting the potential for sustained benefit after a single intervention [61]. It should be noted that this was an innovative study involving avatars to facilitate life review in cancer patients, which concluded high rates of adherence and acceptability, showing that it was feasible and acceptable to carry it out with patients diagnosed with advanced cancer [61]. 

Among TOLF findings [62], the information and exercises provided aimed to enable users to develop self-care skills to manage their lymphedema symptoms, and they were able to improve patient skill management and, consequently, QoL [62]. Themes regarding empowerment, high-quality information, loving the avatar simulation videos, easy accessibility, and user-friendliness were mentioned in patient feedback. Consequently, usability, feasibility, and acceptability of the tested TOLF could contribute to the improvement of the patient QoL [62]. These positive outcomes, together with the high rates of adherence and acceptability of [61], might have been influenced by the effects patients’ motivation or reasons to participate in the studies in the first place, which should be taken into consideration. 

According to the Andersen behavioral model of health services utilization [70,71], reasons to participate could be due to the patients’ predisposing (such as age, gender, or health beliefs), enabling (such as physical accessibility and distance), and need factors related to use of medical care [72]. 

As specified by previous studies [72], these factors can be motivations, intentions, emotional state, needs, and preferences of the patient, and the appeal and perceived value of the intervention, as well as other contingencies related to the intervention (such as demographic data, compensation for participation (usually being monetary), logistics, or effort required by the patient).

Other results confirm that cancer avatars were associated with negative judgments and unpleasant emotions, and appeared as less attractive and self-representative than the AS and IS avatars [63]. Despite that, cancer avatars were no more difficult to create than the others, and they were not necessarily wholly associated with more intense, unpleasant, and difficult-to-deal-with emotions than the AS and IS avatars [63]. Correlations showed that differences in attitudes toward one’s avatars can be related to clinically relevant states: depression and anxiety seemed inversely related to positive attitudes (pleasantness, intensity, and attractiveness) toward the avatars representing AS, suggesting a relationship between depression/anxiety and poor self-worth and the possibility of using avatars to gather information on users’ self-perception [63]. The differences in attitudes towards the different avatars might have also been modulated through the Proteus effect [73], which refers to the phenomenon of the avatar’s appearance influencing the user’s behavior or attitude [74]. This effect is stronger when users can choose and control their avatars [75] and when the self-identification with the avatar (avatar embodiment) is high-er [76].

The study provided insight into the relationship between self-representations (through avatars) and clinically relevant states, thereby empowering medical treatment’s consideration of psycho-cognitive factors [63]. As this was just a descriptive study, the variable of psychological well-being was not modified [63]. 

As previously stated, TOLF [62] was further investigated through a randomized clinical trial (RCT), published in 2022 [77], which concluded there to be significant health benefits for BC survivors (management of chronic pain, soreness, general bodily pain, arm/hand swelling, heaviness, and impaired limb mobility), but it did not show significant differences in mean QoL between the experimental and control groups. These contradictory findings highlight the need to introduce new and more efficient methods of recruitment and stronger RCT designs to figure out why a BC patient’s QoL did not benefit from this technological tool. It is crucial that the main factors providing successful out-comes are properly determined, as well as those factors that could discourage its use in BC patients by measuring different clinical variables (i.e., type of treatment, primary or recurrent diagnosis, physical side effects), or psychological variables (personality, coping style, social support). All the aforementioned points could be useful to establish affordable protocols that allow healthcare staff to check the eligibility of the avatar-based strategies depending on every unique patient’s case.

Considering the outcomes previously discussed, the most successful path to develop avatar-based strategies involving BC patients remains unclear. More evidence-based studies are needed in order to determine stronger conclusions. However, as the Health Euro-pean Union program stressed, the possibility of using a Human Avatar (HA) system, including the Physical Human Avatar (PHA), which can characterize multiple human conditions, is an innovative and real possibility [57]. It should be taken into account in order to achieve genuine and accurate benefits for BC patients via the use of these technological tools. In fact, HAs have recently been mentioned as ideal user interfaces for mobile healthcare applications and biobehavioral feedback for healthy living [57]. As HAs may vary between static images and more dynamic animations, generally giving them humanoid characteristics, such as the ability to act and move [6], they could be customized depending on a patient’s particular clinical case, and even based on the psycho-oncological therapeutic goals considered (patients under active treatment versus survivors, for example). HAs usually maintain aspects of the user’s identity and self-perception, such as their gender, personality traits, physical appearance, social skills, and/or cultural belonging. They can then represent their virtual identity, i.e., a digital self-representation of users in virtual worlds [78,79,80,81].

The use of HAs could be beneficial for improving body image concerns faced by BC patients after a mastectomy, or even to manage disfiguration after hair loss or body weight increase. As one of the included studies reported, HAs could be used for lymphedema prevention, considered as one of the chronic side effects faced by many BC patients. As such, through educational protocols that include HAs, other secondary symptoms could perhaps be addressed in advance, preventing them from reaching high rates of severity, such as sedentary lifestyles or body image distortion. In this regard, cognitive skill training and behaviors that use avatars have been mentioned as unique, since movable shapes could increase user attraction [59]. Moreover, using HAs as an affordable way to deliver education has been stated before, including various types of application systems such as modules, videos, avatars, and features of cultural integration [82,83,84,85]. Avatars have also shown to be promising tools for skill training in children [86], and are a major component of mobile apps in reporting and assessing symptoms to provide appropriate education to young cancer patients [59]. Animal “avatars” have also been implemented in personalized medicine in oncology [87].

Another possibility that HAs could provide BC patients, along with the educational ones, could be reducing the gap between the perceived lack of social and psychological support they mentioned [62]. As HAs could be developed through a smartphone app or a web-based platform, the technological tool could be used in a patient’s home, providing extra help, as well as helping to monitor secondary symptoms or facilitate therapeutic in-formation that could be beneficial for patients, for example, the kind of medication they should keep in mind for managing chronic pain or any clear signal that they should con-sider seeking emergency care. Finally, all these applications could be considered as extra tools for helping BC patients in order to make the whole patient journey that an oncological diagnosis entails more affordable and less painful. 

## 5. Conclusions

In accordance with the objectives proposed and the results obtained in this systematic review, the following conclusions were drawn: 

(1) There is not enough empirical evidence to suggest that avatar-based strategies are an effective technology to enhance BC patients’ QoL or psychological well-being (despite the promising outcomes reported in the analyzed records) due to the scarcity of studies targeting this topic;

(2) Considering the vast heterogeneity of avatar-based applications with BC patients, several strengths, weaknesses, opportunities, and threats should be kept in mind, as well as researched further and more in-depth, in order to properly establish paths for developing this technological tool in the oncological sphere;

There is still a lack of evidence in the literature regarding the usefulness of HAs as educational and therapeutic tools for chronic patients, including women diagnosed with BC [61]. It is imperative to develop accurate studies that allow us a clear understanding of drivers and barriers of HA development.

The authors of this study, as researchers, professors, and healthcare staff, involved at the Psycho-Technology Lab (Universidad San Pablo-CEU, Madrid-Spain), fully support the words of Anneyce Knight, a nurse for 35 years who became a breast cancer patient [88]. As such, we (authors) are “very conscious that the service-user is at the center of the care we all provide and that every service-user is unique”. For that reason, we are strongly encouraged to provide empirically validated tools for helping chronic patients who are dealing with a painful patient journey by considering them as unique and keeping this in mind to provide them with new useful and efficacious therapies (such as avatar-based strategies), as long as they are empirically and ethically validated and patient-centered. If it proves to help reduce the burden on cancer patients, we are all committed to keep working hard in this topic.

### 5.1. Clinical and Researcher Implications

Future randomized controlled trials are needed to determine the following remaining questions. Does the effectiveness of avatar-based strategies depend on the oncological diagnosis? Are there specific types of HA strategies that may produce higher enhancement, engagement, and skill acquisition in BC patients? What are the side effects that could be more successfully addressed through educational protocols involving HAs? Are there any relevant factors that should be taken into account as exclusion criteria regarding HA application in BC patients? Do some psychological variables or specific clinical skills benefit more from the use of HA technologies? Will the improvement be the same using MR as when using, e.g., the Metaverse? Is there any potential harm implied for BC patients after using HA strategies?

### 5.2. Limitations 

The main limitations of this study include the extreme scarcity of studies involving avatar-based strategies in BC patients, as well as the methodological heterogeneity of the studies included. This hinders the uniformity of the results, as well as their generalizability. Because only three studies were finally included in this systematic review and due to they were developed in different countries, and one of them included only one breast cancer patient, the conclusions of these studies must be interpreted with caution. However, in order to produce meta-analyses that provide conclusive results on the effectiveness of this approach, greater homogeneity in the targeted patients as well as in the methodology used would be desirable, especially involving breast cancer patients [89]. Similarly, other kinds of reviews, such as literature or scoping reviews, could be helpful in increasing the knowledge regarding the usefulness of avatar-based strategies for breast cancer patients. More descriptive and inclusive reviews could also provide rich information, such as those including cancer patients’ self-representations with multiple tools, until more specific avatar-based methods emerge and can be tested through RCT. 

Likewise, the devices that deploy avatars are undergoing significant and rapid advancement, such as Metaverse One, so studies carried out regarding their advantages and disadvantages should be updated and renewed according to the latest technological developments. 

## Figures and Tables

**Figure 1 cancers-15-04031-f001:**
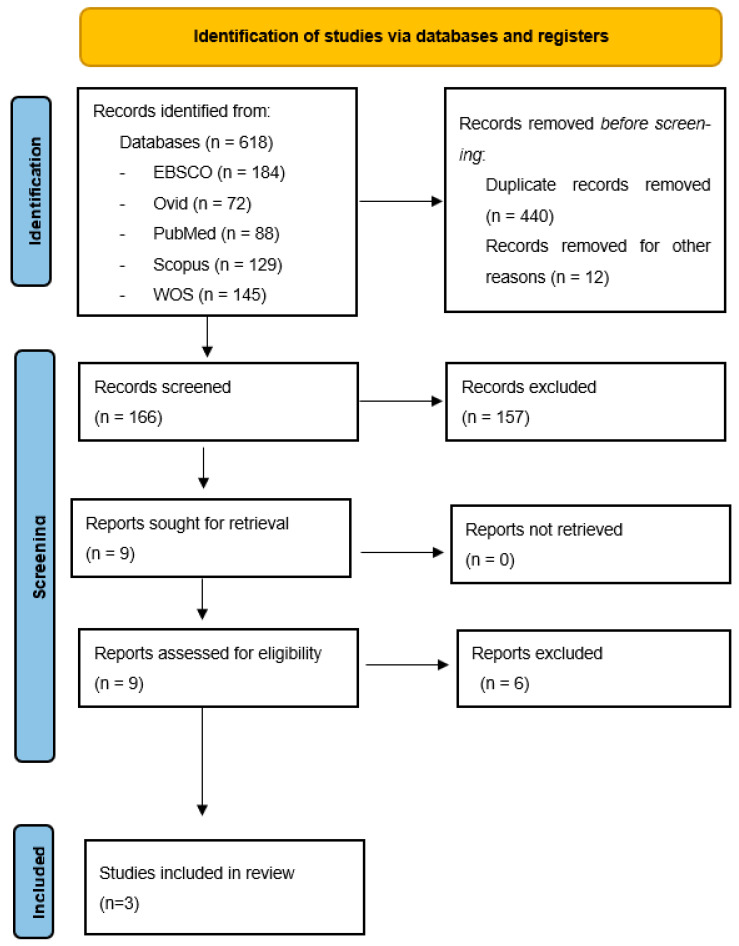
Systematic review of the literature flowchart.

**Table 1 cancers-15-04031-t001:** General characteristics of the included studies (n = 3).

Study	Publication Year	Country	Study Design
Dang et al. [61]	2021	USA	Mixed method
Fu et al. [62]	2016	USA	Mixed method
Triberti et al. [63]	2019	Italy	Quantitative

**Table 2 cancers-15-04031-t002:** General characteristics of the included studies (II) (n = 3).

Study	Aim	Sample Size (Mean Age)	Only Breast Cancer Patients/Stage	Patients Randomized
Dang et al. [61]	Establish the feasibility and acceptability of an avatar-facilitated life review intervention.	12 (24–65+ years)	No/NP	No
Fu et al. [62]	Test the TOLF system to evaluate the reliability, validity, and efficacy of mHealth assessment as well as usability, feasibility, acceptability, and efficacy of mHealth self-care interventions for lymphedema symptoms among the end user of breast cancer survivors.	355 (21–80 years)	Yes/NP	No
Triberti et al. [63]	Examine how breast cancer affects avatar-conveyed multiple representations and if attitudes toward avatars are related to anxiety and depression.	22 (49.4)	Yes/NP	No

NP = No provided by authors.

**Table 3 cancers-15-04031-t003:** Primary outcomes (n = 3).

Study	Training Using Avatars	QoL ^c^ and Well-Being Assessment	Useful to Improve QoL/Well-Being
Dang et al. [61]	VoicingHan via a Perception Neuron ^a^ and a Logitech^®^ wireless headset	Yes EORTC QLQ-C30 ^b^, FACIT-Sp ^c^, ESAS ^d^	No/Yes
Fu et al. [62]	TOLF ^e^ (app and website)	Yes BCLE-SEI ^f^	Yes/Yes
Triberti et al. [63]	Profile Avatar Maker 2 (app)	No/Yes PHQ-9 ^g^, GAD-7 ^h^	Not applicable

^a^ Perception Neuron: it is a Motion Capture^®^ (MoCap) device; ^b^ EORTC QLQ-C30: European Organization for Research and Treatment of Cancer—Quality of Life Questionnaire Core 30; ^c^ FACIT-Sp: Spiritual Well-Being Scale; ^d^ ESAS: Edmonton Symptom Assessment System; ^e^ TOLF: The Optimal Lymph Flow; ^f^ BCLE-SEI; ^g^ PHQ-9: Depression Patient Health Questionnaire-9; ^h^ GAD-7: Anxiety Generalized Anxiety Disorder-7.

**Table 4 cancers-15-04031-t004:** Secondary outcomes (n = 3).

Study	Main Advantages	Main Disadvantages	Patients’ Satisfaction
Dang et al. [61]	High feasibility; avatars in VR enhance patient communication and engagement, and promote the expression of identity	Selection bias; intervention requires space and technical expertise	High
Fu et al. [62]	Patient-targeted; usability tested; feasible, user-friendly; avatar tailored to the specific population	Homogeneous sample in some characteristics	High
Triberti et al. [63]	The experience of a chronic illness (cancer) was used as an independent source of information to represent a facet of the self. Patient´s privacy preserved.	Lack of measurement in avatar characteristics and in the ability and satisfaction of the participants with their avatar creation	Unknown

## Data Availability

The data can be shared up on request.

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
