# Peer review of "Avatar-Based Strategies for Breast Cancer Patients: A Systematic Review"

_cancers, 2023, doi:10.3390/cancers15164031_

Round 1

Reviewer 1 Report (New Reviewer)

This is an interesting attempt to review scientific literature on avatars involved in cancer management. The topic is very innovative and the review is well performed, although I think Authors should make some other efforts to improve both precision and appropriateness of reporting the reviewed studies

- the third study in table 1 is deemed "qualitative" while it uses quantitative analyses - there is no qualitative research in that study so this should be corrected

- the second study is described throughout the review in its many complex aspects but the specific role of avatars, and how were they managed within the study and the application, is not completely clear - the description should focus on the topic of the review

- when such a low number of papers are found to review, Authors are expected to reflect on whether a systematic review was the correct approach to begin with. As I said I think the final review product is convincing but we still have a systematic review with only 3 studies quite different from each other. Authors in the end prefigure a future meta-analysis, but I think that the conclusion should say that more descriptive and inclusive reviews (e.g., literature or scoping reviews) should be encouraged, until specific avatar based methods emerge and could be effectively compared - in other words, I would suggest for future research directions to encourage inclusive, wider reviews to capture more general exclusion criteria (e.g.: studies on cancer patients' self-representations with multiple tools, avatars included). 

Again, I think the present review deserve consideration - but sometimes review conclusions should tell you that you have to go back and re-open your gaze, not close it further 

- in some parts of the review Authors talk about "avatar therapies" but I think the studies (and the role of avatars in them) could be hardly considered "therapeutic", possibly more based on assessment and basic research - I would correct this aspect or try to explain more clearly what Authors meant

- references numbering should be revised - apparently some references were added and the numbering does not mirror the text

Author Response

Dear reviewer.

We do really appreciate your kind report about our manuscript.

Please find attached the point-by-point response to each comment suggested.

Thank you so much for taking the time to revise this manuscript and consequently, helping us for its significative improvement.

Kind regards,

Authors.

Reviewer 2 Report (New Reviewer)

Dear authors

thank you for this very interesting review on Avatar based therapy for breast cancer patient. Your review covers a clinical urge in rural circumstances as well as in a 'anonymous' city life and helps clinicians to make a sound decision. With your clear description of the method (PRISMA) anyone interested can easily repeat your search. 
the limitations are  clearly pointed out and future research fields identified. Your conclusion sums the review up understandable. 
the only minor thing would be a native speaker  correction - but me being a non native speaker myself, this might just be my ‘dialect’. 

See above

Author Response

Dear reviewer.

We do really appreciate your kind report about our manuscript.

We will apply for extensive English editing service in MDPI.

Thank you so much for taking the time to revise this manuscript and consequently, helping us for its significative improvement.

Kind regards,

Authors.

This manuscript is a resubmission of an earlier submission. The following is a list of the peer review reports and author responses from that submission.

Round 1

Reviewer 1 Report

Manuscript ID: cancers-2133203, Title: Avatar-based Therapy for Breast Cancer Patients: A Systematic Review submitted by Irene Rodríguez-Guidonet has discussed recent topic for cancer therapy. Upon reading the manuscript I have obsorved few points needs to rectify while addressing revision.   1. On the 1st and 2nd page there are many grammatical errors which needs to rectify. 2. In my opinion in the materials & methods- selection criteria - may be not needed for exclusion criteria please confirm. 3. In training using Avatar- there are no data about how many breast cancer patients are there. 4. Are patients taken randomly, like any cancer patients? 5. Not also given the data that which stage the disease like stage -1,2,3 or 4, the patients which are got advantages after the avatar-based therapy.  

Reviewer 2 Report

This paper shows the paucity of research in using Avatar to treat the stressors associated with a diagnosis of breast cancer. This was acknowledged by the authors.

There are 3 papers cited in this paper but one of them (Dang) has only 1 patient that had breast cancer. I am not sure how much conclusion can be drawn from one patient and I am not sure if it should be included in the analysis. 

Citation #5 may not be appropriately referenced on the second page. 

Reviewer 3 Report

Dear Authors,

thank you for this interesting contribution. Here are some of my suggestions;

INTRODUCTION

 - Addressing eHealth and Avatar, it is relavant to manage Body Image and its related bodily issues. Please, make Introduction consistent with more details about them (see: Boquiren et al.,., 2013; JabÅ‚oÅ„ski ET AL., 2019)

 - As mentioned, social support is fundamental. Please, clarify this point in reference to your focus of research interest (see: Martos-Méndez  et al., 2015; Sebri et al., 2021)

 - To make the background more consiste, please provide a definition of eHealth and Avatars as well as their prons and cons. Specifically, be carefull to oncological patients and ther virtual Body Image

 - Similarly, make also objectives cleares. Why do you evaluate that this study could be helpful for the current literature? Why for women with a breast cancer?

MATERIALS AND METHOD

 - The research was carried out in November, 2022. Explain why did you not update your research/update it.

 - I am not sure that the following keywords as “avatar” + “cancer”; and “avatar” + “breast cancer” is sufficient to cover all the existing literature. Please, motive your choice.

- Please, add the "risk of bias " table

RESULTS

 - Only three articles are included a the end. I argue that it is not sufficient to review this topic of interest. maybe it could be better to do the research another time or change something over research process.

DISCUSSION

- Please, make the discussion section more consistent with the current literature and studies about avatars (e.g. Prometheus effect and so on)

- Participants motivation to participate in a study could make the difference, expecially in virtual conditions. Please, add this point in your discussion/conclusion (see: Savioni et al., Savioni et al., 2022)

REFERENCES

Boquiren, V.M.; Esplen, M.J.; Wong, J.; Toner, B.; Warner, E. Exploring the influence of gender-role socialization and objectified body consciousness on body image disturbance in breast cancer survivors. Psycho-Oncology 2013, 22, 2177–2185.

JabÅ‚oÅ„ski, M.J.; Mirucka, B.; Streb, J.; SÅ‚owik, A.J.; Jach, R. Exploring the relationship between the body self and the sense of coherence in women after surgical treatment for breast cancer. Psycho-Oncology 2019, 28, 54–60.

 Martos-Méndez M. J. (2015). Self-efficacy and adherence to treatment: the mediating effects of social support. J. Behav. Health Soc. Issues 7, 19–29. 10.5460/jbhsi.v7.2.52889

Savioni, L., Triberti, S., Durosini, I., Sebri, V., & Pravettoni, G. (2022). Cancer patients’ participation and commitment to psychological interventions: a scoping review. Psychology & Health, 37(8), 1022-1055.

Sebri, V., Mazzoni, D., Triberti, S., & Pravettoni, G. (2021). The impact of unsupportive social support on the injured self in breast cancer patients. Frontiers in Psychology, 12.

Reviewer 4 Report

Although only three studies were finally analyzed in the manuscript, I think it was a valuable review and would have contributions to future related research. A typo error in Abstract, “…Scopus y WOS (Web of Science)…” can be amended. Besides, I have no other comment.

Author Response

Dear reviewer.

We do really appreciate your kind report about our manuscript.

Thanks a lot for providing us this consideration. We have corrected the typo in the abstract (page 1).

Kind regards,

Authors.